# Improved pearl millet genomes representing the global heterotic pool offer a framework for molecular breeding applications

Punna Ramu [1], Rakeh K. Srivastava [2✉], Abhijit Sanyal[1], Kevin Fengler[3], Jun Cao[3], Yun Zhang [3], Mitali Nimkar[1], Justin Gerke[3], Sriram Shreedharan[3], Victor Llaca [3], Gregory May[3], Brooke Peterson-Burch [3], Haining Lin[3,5], Matthew King[3,6], Sayan Das[1], Vaid Bhupesh[1], Ajin Mandaokar[1], Karunakaran Maruthachalam [1], Pobbathi Krishnamurthy[1], Harish Gandhi [2,7], Abhishek Rathore [2,8], Rajeev Gupta[2,9], Annapurna Chitikineni[2,10], Prasad Bajaj[2], S. K. Gupta [2], C. Tara Satyavathi[4], Anand Pandravada[1], Rajeev K. Varshney [2,10✉] & Raman Babu [1✉]

High-quality reference genome assemblies, representative of global heterotic patterns, offer an ideal platform to accurately characterize and utilize genetic variation in the primary gene pool of hybrid crops. Here we report three platinum grade *de-novo*, near gap-free, chromosome-level reference genome assemblies from the active breeding germplasm in pearl millet with a high degree of contiguity, completeness, and accuracy. An improved Tift genome (Tift23D$_2$B$_1$-P1-P5) assembly has a contig N50 ~ 7,000-fold (126 Mb) compared to the previous version and better alignment in centromeric regions. Comparative genome analyses of these three lines clearly demonstrate a high level of collinearity and multiple structural variations, including inversions greater than 1 Mb. Differential genes in improved Tift genome are enriched for serine O-acetyltransferase and glycerol-3-phosphate metabolic process which play an important role in improving the nutritional quality of seed protein and disease resistance in plants, respectively. Multiple marker-trait associations are identified for a range of agronomic traits, including grain yield through genome-wide association study. Improved genome assemblies and marker resources developed in this study provide a comprehensive framework/platform for future applications such as marker-assisted selection of mono/oligogenic traits as well as whole-genome prediction and haplotype-based breeding of complex traits.

[1] Corteva Agriscience, Hyderabad, Telangana, India. [2] International Crops Research Institute for the Semi-Arid Tropics, Hyderabad, Telangana, India. [3] Corteva Agriscience, Johnston, IA 50131, USA. [4] Indian Council of Agricultural Research – All India Coordinated Research Project on Pearl Millet, Jodhpur, India. [5]Present address: Moderna, 200 Technology Square, Cambridge, MA 02139, USA. [6]Present address: Natera Inc, San Carlos, CA 94070, USA. [7]Present address: International Maize and Wheat Improvement Center (CIMMYT), Nairobi, Kenya. [8]Present address: International Maize and Wheat Improvement Center (CIMMYT), Hyderabad, India. [9]Present address: Cereal Crops Research Unit, Edward T. Schafer Agricultural Research Center, USDA-ARS, Fargo, ND 58102, USA. [10]Present address: Centre for Crop & Food Innovation, State Agricultural Biotechnology Centre, Food Futures Institute, Murdoch University, Murdoch, WA 6150, Australia. ✉email: r.k.srivastava@CGIAR.ORG; rajeev.varshney@murdoch.edu.au; raman.babu@corteva.com

Ensuring food security for the burgeoning world population, especially in the wake of climate change and associated adverse effects, is a major challenge that necessitates approaches including crop diversification and targeted improvement. Significant advances have been made to improve grain yields in major cereals such as maize, wheat and rice in recent decades, whose primary adaptation however continues to be well-endowed agricultural ecologies. Pearl millet [*Pennisetum glaucum* (L.) R. Br., syn. *Cenchrus americanus* (L.) Morrone], a $C_4$ grass ($2n = 2x = 14$), adapted to harsher environments in the arid and semi-arid regions of the world accounts for roughly 50% of the world's millet production and is considered a boon to geographies that are vulnerable to climate change and prone to depleting water resources and widespread malnutrition[1]. Despite impressive progress made with respect to developing and deploying high-yielding cultivars by public and private sector partnerships in addition to improved agronomic management practices, the productivity of pearl millet continues to be as low as ~1.2 t/ha[2,3]. Molecular breeding along with other associated strategies such as speed breeding, high throughput phenotyping and breeding informatics will likely accelerate future genetic gains.

A pioneering effort by the International Crops Research Institute for the Semi-Arid Tropics (ICRISAT) resulted in a reference genome assembly (referred to as Tift-2017[4]) based on the inbred line, Tift23$D_2$$B_1$-P1-P5, released in 2017. While this ushered the very first draft genome for the pearl millet research community, the shotgun sequencing and BAC sequencing approaches resulted in a highly fragmented assembly with thousands of contigs and numerous gaps that posed considerable challenges in gaining a comprehensive understanding of genomic organization and evolution for breeding applications. In this study, we report platinum-grade de novo genome assemblies with high-quality annotation using high-coverage long and short read data for three pearl millet genotypes, including an improved genome assembly for the previously published Tift23$D_2$$B_1$-P1-P5. The two other genotypes (843 B and ICMR 06777) were carefully selected to represent the global heterotic pattern as established by the ICRISAT pearl millet breeding program. Inbred line 843 B is an A1 cytoplasm[5] maintainer line while ICMR 06777 is an A1 cytoplasm restorer line. Both 843 B and ICMR 06777 represent the current hybrid parents' gene pool of the Asian pearl millet breeding programs. The maintainer line 843 B represent the maintainer (B-) line breeding program. It is one of the key founder lines for the B- line gene pools that have been bread so far. The restorer (R-) line ICMR 06777 represents the restorer breeding pools.

## Results and discussion

### Genome assembly of inbred lines.
The three new pearl millet genomes were assembled using PacBio HiFi sequencing, de novo contig assembly, and hybrid scaffolding with Bionano optical maps (See Methods). The improved Tift genome (referred as Tift) was assembled into 46 contigs with a contig N50 of 126.5 Mb and scaffold N50 of 144 Mb (Table 1) whereas previously published Tift-2017 exhibited a contig N50 of 0.018 Mb and scaffold N50 of 0.88 Mb[4]. Two other genomes, 843 B and ICMR 06777, were also assembled using the same approach as the Tift genome and produced scaffold N50s of 144.6 Mb and 279 Mb, respectively (Table 1). A total of 24 scaffolds and 23 scaffolds were assembled into 7 pseudomolecules in Tift and 843 B, respectively. Comparatively, ICMR 06777 assembly is captured in just ten scaffolds, of which, five chromosomes were spanned by a single scaffold (Chr2, Chr3, Chr5, Chr6, and Ch7), chromosome 1 was covered by two scaffolds, and chromosome 4 was spanned by three scaffolds. All three genome assemblies (Tift, ICMR 06777 and 843

B) are more contiguous and larger in size relative to Tift-2017 (Table 1). The estimated GC content rose slightly higher in the improved Tift assembly (49.1%) compared to Tift-2017 (47.9%).

Given the high level of contiguity and completeness, genome assemblies are an important genomic resource for trait discovery and deployment, and particularly for pangenomic, haplotypes-based and introgression breeding[6]. To estimate the genome contiguity, we identified the density of gaps in the new assemblies and compared them with Tift-2017 (Supplementary Fig. 1, Table 1). The ICMR 06777 assembly contains the least gaps and represents one order of magnitude improvement in contiguity over the other genomes. Tift-2017 genome has 6432 gaps larger than 100 bp compared to 25 gaps in the new Tift assembly, 3 gaps in ICMR 06777 and 32 gaps in 843 B. The most distinguishing characteristic among the three pearl millet assemblies is that the PacBio contigs from ICMR 06777 better captured centromeric regions, leading to more complete hybrid scaffolds (see Methods). As highlighted by previous studies[7,8], advances in sequencing chemistry and more efficient assembly algorithms may have played a significant role in achieving improved contiguity and better centromere placements in the current study.

Having comparable high-quality genome assemblies from three pearl millet lines, constructed using the same genomics technologies, enables pangenome comparisons where true sequence and structural variations such as indels and copy number variations can be distinguished from artifacts arising from different methodologies. TagDot plots (see Methods) were constructed for all assemblies to facilitate genome-scale comparisons and to assess assembly quality, particularly between assemblies from the same genotype constructed with different methods. These plots highlighted the high degree of collinearity between these genomes punctuated with 15 (Tift) and 9 (843 B) inversions greater than 1 Mb compared to ICMR 06777 assembly (Fig. 1, Supplementary Fig. 2). For example, ICMR 06777 exhibits an 8.75 Mb inversion on chromosome 4 relative to the 843 B and Tift genomes (Supplementary Fig. 2). Interestingly, this inversion harbors ~50 genes, including disease resistance genes and the *dehydrin* gene (one copy in Tift and 843 B, and two copies in ICMR 06777, Supplementary Data 1) which has been demonstrated to play a significant role for drought tolerance in wheat[9]. Contiguity is increased in new assemblies and significantly better alignments were observed in the centromere regions in comparison to Tift-2017 assembly (Fig. 1, Supplementary Fig. 2). The chromosome-scale assembly approach adopted in the current study revealed higher levels of genomic synteny (78%, Fig. 2) among the newly assembled genomes. Structural variations like inversion were observed on chromosome 3 (red arrows) in all new assemblies, while chromosome 2 and 3 shared duplicated genomic segments (black arrows) (Fig. 2). In addition, the distribution of identical genomic segments between chromosome 1 and 6 (blue arrows) suggested a role for paralogous genes causing the secondary syntenic relationships, thereby lending credence to the paleopolyploid origin of pearl millet genome.

### Genome annotation.
Genome annotation is essential to understand the functional and non-functional segments of a genome. To identify protein coding genes and repeat regions, three millet genomes were annotated with a hybrid approach comprising de novo gene prediction and evidence-based methods. Above analysis resulted in a set of 35,000–38,000 genes (non-TE related, and supported by transcript, PFAM and/or Uniprot evidence), of which 500–700 were tRNA genes and the rest being protein-coding genes accounting for nearly 7% of the genome (Table 1). Long reads generated using PacBio IsoSeq v3 on transcriptome libraries from the three genotypes were instrumental in predicting

**Table 1 Summary statistics of pearl millet genome assemblies.**

| | ICMR 06777 | 843 B | Tift | Tift-2017 |
|---|---|---|---|---|
| *Assembly* | | | | |
| Sequencing method | PacBio HiFi | PacBio HiFi | PacBio HiFi | Illumina WGS/BAC |
| Assembly method | hifasm | hifasm | hifasm | multiple |
| Scaffolding method | BioNano mapping | BioNano mapping | BioNano mapping | multiple |
| PacBio sequencing coverage | 29x | 22x | 39x | N/A |
| Minimum PacBio Read Length (kb) | 9 | 8 | 7 | N/A |
| Minimum Contig Coverage | 15x | 5x | 20x | N/A |
| Minimum contig length (kb) | 60 | 70 | 70 | N/A |
| PacBio HiFi Read Length N50 (kb) | 15.4 | 15.6 | 17.4 | N/A |
| Contigs | 88 | 95 | 46 | 175,708 |
| Contig N50 (Mb) | 129.4 | 71.7 | 126.5 | 0.018 |
| Total contig length (Gb) | 1.933 | 1.866 | 1.859 | 1.556 |
| QV | 55.79 | 54.95 | 53.8 | N/A |
| BioNano effective molecule coverage | 61.3 | 90.7 | 99.3 | N/A |
| BioNano molecule N50 (kb) | 353.8 | 405.2 | 443.0 | N/A |
| BioNano maps (>1 Mb) | 51 | 32 | 33 | N/A |
| BioNano map N50 (Mb) | 104.3 | 139.6 | 143.5 | N/A |
| Total map length (Mb) | 1855.8 | 1851.4 | 1869.1 | N/A |
| Scaffolds | 10 | 23 | 24 | 25,241 |
| Scaffold N50 (Mb) | 279.0 | 144.6 | 144.0 | 0.885 |
| Pseudomolecule length (Gb) | 1.922 | 1.860 | 1.859 | 1.540 |
| Pseudomolecule Gaps (spanned) | 24 | 51 | 37 | 133,441 |
| Pseudomolecule Gaps (unspanned) | 3 | 32 | 25 | 6,432 |
| *Annotation* | | | | |
| Number of genes | 50339 | 57822 | 54176 | 37643 |
| Number of protein-coding genes | 35036 | 38069 | 36510 | 35791 |
| Number of tRNA genes | 532 | 705 | 713 | 879 |
| Number of mRNAs | 58078 | 65228 | 62882 | 35791 |
| Number of exons | 213705 | 217957 | 224575 | 147370 |
| Number of introns | 170994 | 172532 | 179403 | 111579 |
| Number of CDS | 42671 | 45393 | 45141 | 35791 |
| Overlapping genes | 961 | 984 | 1021 | 155 |
| Contained genes | 133 | 145 | 143 | 58 |
| Average gene length (bp) | 3329 | 3214 | 3286 | 2438 |
| Average mRNA length (bp) | 3534 | 3412 | 3514 | 2438 |
| Average exon length (bp) | 300 | 302 | 302 | 248 |
| Average intron length (bp) | 509 | 519 | 509 | 456 |
| Average CDS length (bp) | 1197 | 1166 | 1185 | 1021 |
| % of genome covered by genes | 6.7 | 7.5 | 7.3 | 5.6 |
| % of genome covered by CDS | 3 | 3.3 | 3.3 | 2.3 |
| mean mRNAs per gene | 1 | 1 | 1 | 1 |
| mean exons per mRNA | 5 | 5 | 5 | 4 |
| mean introns per mRNA | 4 | 4 | 4 | 3 |

a ~1 kb increase in average gene model length compared to Tift-2017. Further, average gene, exon and CDS length were ~1 kb, ~50 bp and ~150 bp longer, respectively, in the new assemblies compared to Tift-2017. Interestingly, ICMR 06777 had the lowest number of rRNA, and tRNA genes compared to the rest of the assemblies (Supplementary Table 1). Gene cluster analysis suggested that most genes fall under core clusters (34.7%) and their duplicates (32.4%) (Supplementary Fig. 3). The remaining clusters were categorized into variable (showing presence absence variation, PAV), singleton (line-specific genes present at a single locus) and duplicated (line-specific genes duplicated at multiple loci) clusters. ICMR 06777 possessed the highest number of line-specific singleton genes (Supplementary Fig. 3). Singletons are rare variants that offer insights into past demographic events in a species, when studied across geographical space. By simulating the empirical distribution of singletons, ref. [10] obtained results supporting West Africa (Northern Mali) as most probable origin of geographic expansion for the cultivated species of Pearl Millet. Interestingly, the higher frequency of singletons as discovered in the current study in ICMR 06777 corroborates the above finding

as the line is derived from the founder population, MC94 which is a mixture of Iniari landraces from Togo, Ghana and other West African countries.

A reciprocal BLAST hit search found that ~94% of Tift-2017 predicted genes were shared with the improved Tift assembly. This indicates that the improved Tift assembly captures the most protein coding regions identified in Tift-2017, while improving significantly on gene structure, gene model completeness and discovering the alternative transcripts. Compared to Tift-2017, 2855 novel genes were identified in the improved Tift assembly, of which 790 genes had Pfam domain and GO annotations. These novel genes are enriched for serine O-acetyltransferase activity and glycerol-3-phosphate (G3P) metabolic processes (Supplementary Fig. 4). Serine O-acetyl-transferase plays a key role in biosynthesis of sulfur-containing amino acids like methionine and cysteine amino acids, which increases the nutritional quality of protein in the seed[11]. Most of the coarse cereal grains (such as maize) as well as legumes are deficient in essential amino acids such as methionine and cysteine, rendering their proteins of lower quality. In contrast,

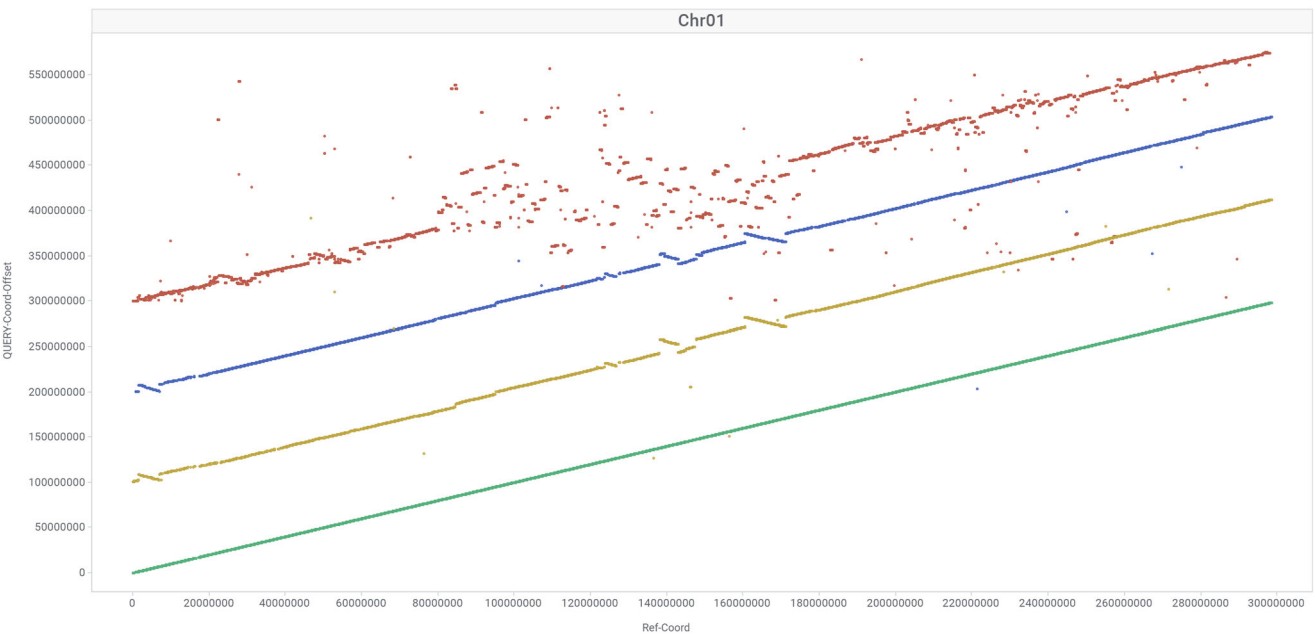

**Fig. 1 Comparison of three millet genome assemblies–Chromosome 1.** Comparing the genome assemblies of Tift (green), ICMR 06777 (yellow), 843 B (blue) and Tift-2017 (red). TagDots (see methods) highlights the structural variation among three genomes and the improvement in the pericentric region relative to the previous reference genome.

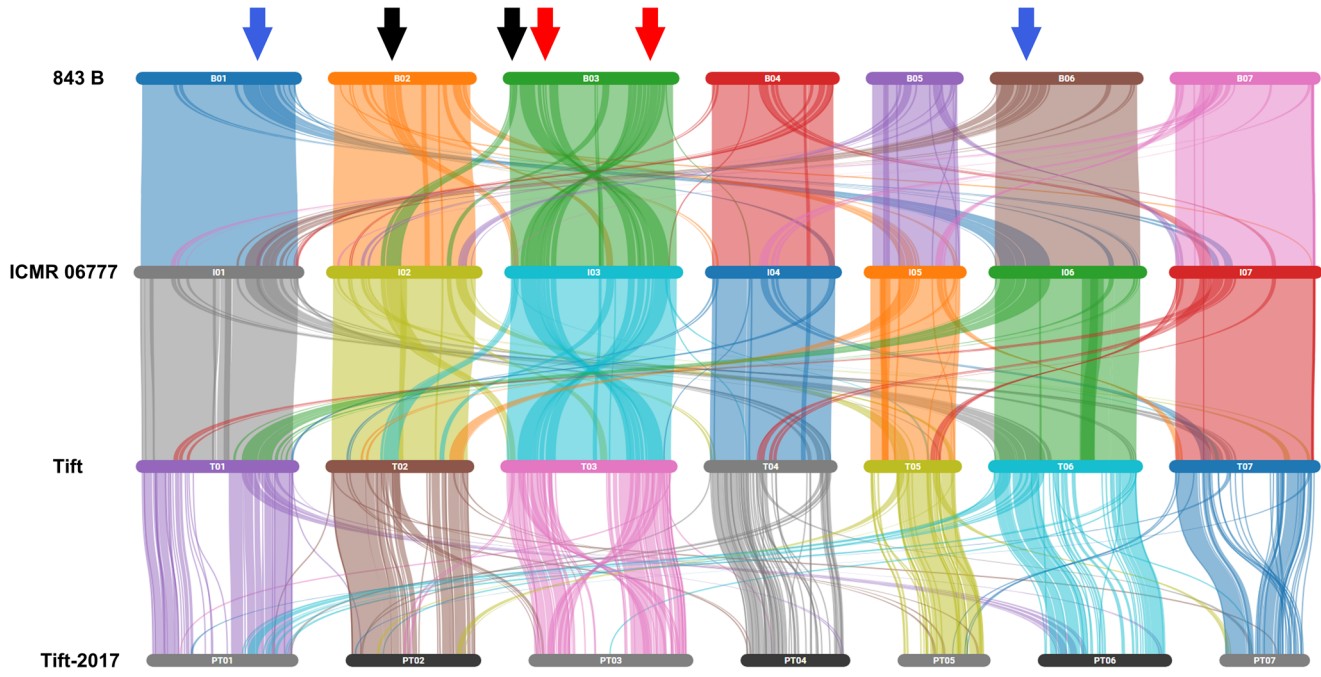

**Fig. 2 Genome synteny and collinearity of millet genomes.** Red arrows = inversion, Black and blue arrows inter-chromosomal segmental duplication. Breaks in synteny in Tift-2017 line is due to draft level assembly. Green arrow is 8.5 Mb inversion on Chromosome 4.

pearl millet contains almost 50% higher concentrations of sulfur-containing amino acids such as cysteine and methionine[11] and high concentrations of micronutrients, namely Fe and Zn, making it a balanced dietary choice to fight against malnutrition[12]. Accumulation of increased G3P has been shown to enhance resistance to fungal diseases[13].

Benchmarking Universal Single-Copy Orthologs (BUSCO) gene coverage is an indication of annotation quality in terms of completeness of single-copy orthologs. BUSCO's coverage (v5.1.2)[14] in the new assemblies is significantly improved (92%) in comparison to Tift-2017 (78%) (Supplementary Fig. 5). Once

again, long reads played a significant role in achieving higher coverage and a substantial reduction in the number of fragmented BUSCOs in the new assemblies. Missing BUSCO genes may be due to the sampled organism state, perhaps missing tissue, or time specific genes. Annotation Edit Distance (AED)[15] values for all new assemblies were <0.5 for >90% of the transcripts (Supplementary Figs. 6–7). Further, ~60%, ~65% and ~74% of the transcripts in Tift, ICMR 06777 and 843 B, respectively, had an AED <= 0.25. This indicated that most transcripts were well supported by the available evidence and the requirement for manual curation was minimal. Less than 2% of gene models had

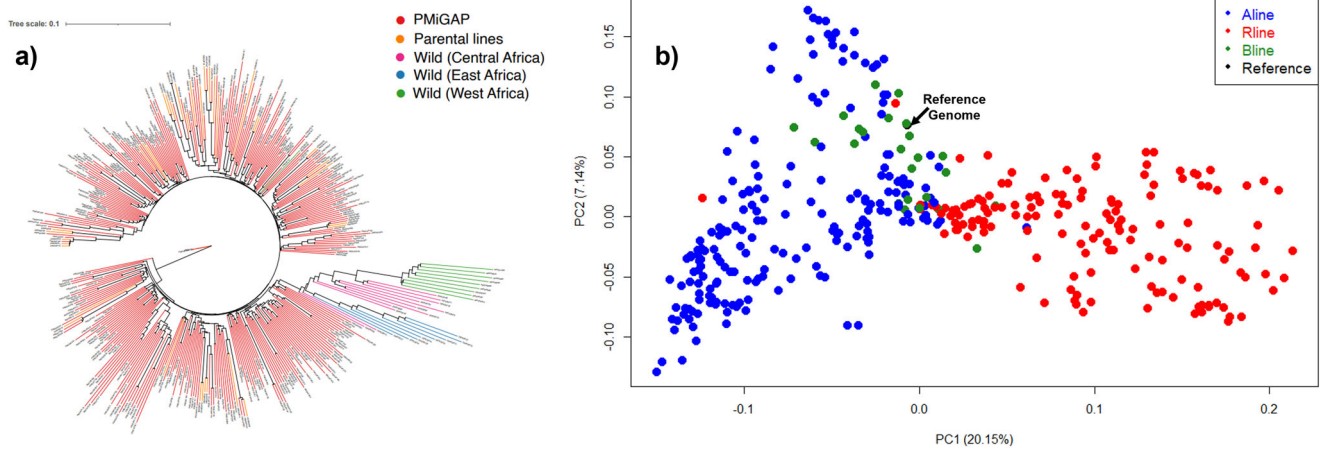

**Fig. 3 Deploying improved reference genome in identifying population structure in millet. a** Genetic diversity analysis of 1028 germplasm lines using SNPs identified on new Tift genome. Grouping of the lines agrees with previous study. **b** PCA plot shows clear separation of heterotic gene pools in Corteva breeding germplasm.

insufficient support and were therefore of low-quality. Classification of the genome assemblies in terms of LTR assembly index (LAI)[16] indicated that all three assemblies were of platinum-grade with indices higher than 25, which agrees with the improved genome assemblies of maize[7,17]. In contrast, the Tift-2017 assembly was of draft quality with the adjusted LAI at 3.70. A substantial portion of the genome sequence is accounted for repeat units. To understand the repeat landscape in the genome, repeats were predicted with Poales interspersed repeats from Repbase[18]. Classification of interspersed repeats indicated that LTR-retrotransposons (LTR-RT) dominated the genome landscape accounting for between 46–51% of the genomic sequence (Supplementary Fig. 8). The majority of LTR-RT account for Gypsy type. The percentage of LTR-RT in new assemblies is higher than the Tift-2017, while the genomic landscape without interspersed repeats is considerably lower in new assemblies (Supplementary Fig. 8).

**Comparative genome analysis.** Many cereal crop species' genomes derive from their last common ancestor, resulting in chromosomal synteny and collinearity among them. Chromosomal rearrangements in pearl millet were evaluated by aligning the improved Tift genome assembly to the maize B73 v5.0[19], rice vJaponica7.1[20], sorghum BTx623 v3.1[21], and foxtail millet[22] genomes. Using annotated genes as anchors, we conducted ortholog analysis for genome segments between Tift and maize B73. Evolutionary conservation between the two species is illustrated by the pairwise chromosomal comparison. Dual alignment locations for most of Tift sequences on maize B73 sequences (Supplementary Fig. 9a) revealed the ancient maize whole genome duplication event[23]. As shown in Supplementary Fig. 9 (b–d), Tift shows contiguous synteny lines between rice, sorghum, and foxtail millet, highlighting the chromosomal level conservation among the species.

**Deploying new reference genomes in millet breeding.** Genetic improvement depends on the extent of the germplasm diversity available in a species. To understand the genetic diversity underpinning the complex and rich morphological diversity in millet populations, resequencing data[4] from 1028 individuals was aligned to the improved Tift genome of the current study which resulted in the discovery of 14.8 million SNPs. A filtered sub-set of 646,328 SNPs (see methods) was used in the PCA-based cluster

analysis which clearly identified the wild ancestors from cultivated germplasm (Fig. 3a, Supplementary Fig. 10) as well as distinguished multiple sub-populations within the primary gene pool. Analysis of independent breeding germplasm at Corteva Agriscience™ identified a clear separation of two heterotic pools available in millet (Fig. 3b). Further, a subset of SNPs was added onto a multi-crop Infinium XT marker platform at Corteva Agriscience™, which is routinely used in whole genome prediction approaches to create targeted hybrids. Higher prediction accuracy was observed for different traits in millet: 0.73 for grain yield (Fig. 4a), 0.94 for flowering time (Fig. 4b) and 0.90 for plant height (Fig. 4c).

Genome-wide association analysis (GWAS) analysis was carried out using the phenotypes generated on breeding germplasm at Corteva and identified marker-trait associations for downy mildew resistance which is colocalized with previously identified QTL in pearl millet[24]. Using three season's phenotypic data sets under control, early and late drought conditions at ICRISAT'S precision field, a GWAS analysis was also performed on the diversity panel representing global genetic diversity of pearl millet (PMiGAP[4]) using a subset of 105,747 SNPs discovered in the current study and identified ~100 significant marker-trait associations (MTAs) for six agronomic traits (Supplementary Data 2) including test grain weight (Fig. 4d). In silico analysis identified six genes differing between male and female genetic pools (Supplementary Table 2) which are the potential targets for fertility restoration in pearl millet. These genes showed homology to earlier reported *Rf* genes from sorghum (*Sorghum bicolor*)[21], sunflower (*Helianthus annuus*)[25], Setaria (*Setaria viridis*)[26], and maize (*Zea mays*)[27].

Haplotype characterization helps the breeders to better understand the underlying allelic variation at the locus level. In the current study, high coverage germplasm lines (448 lines[4], Supplementary Fig. 11a) were used to identify the haplotypes for all 36,510 genes identified in the improved Tift assembly. Of which, haplotype groups were built for 35,858 gene models with at least 20 SNPs in the gene region with 100 KB flanking sequence (Supplementary Fig. 11b). Major haplotypes averaged at 3.6 groups, and peaked at a median of 3 groups, where a major group contained at least 10 lines. The number of haplotype groups ranged from 1 to 14 in the population (Supplementary Fig. 11c). This information is of potential use to the breeders when superior haplotypes could be identified through well characterized

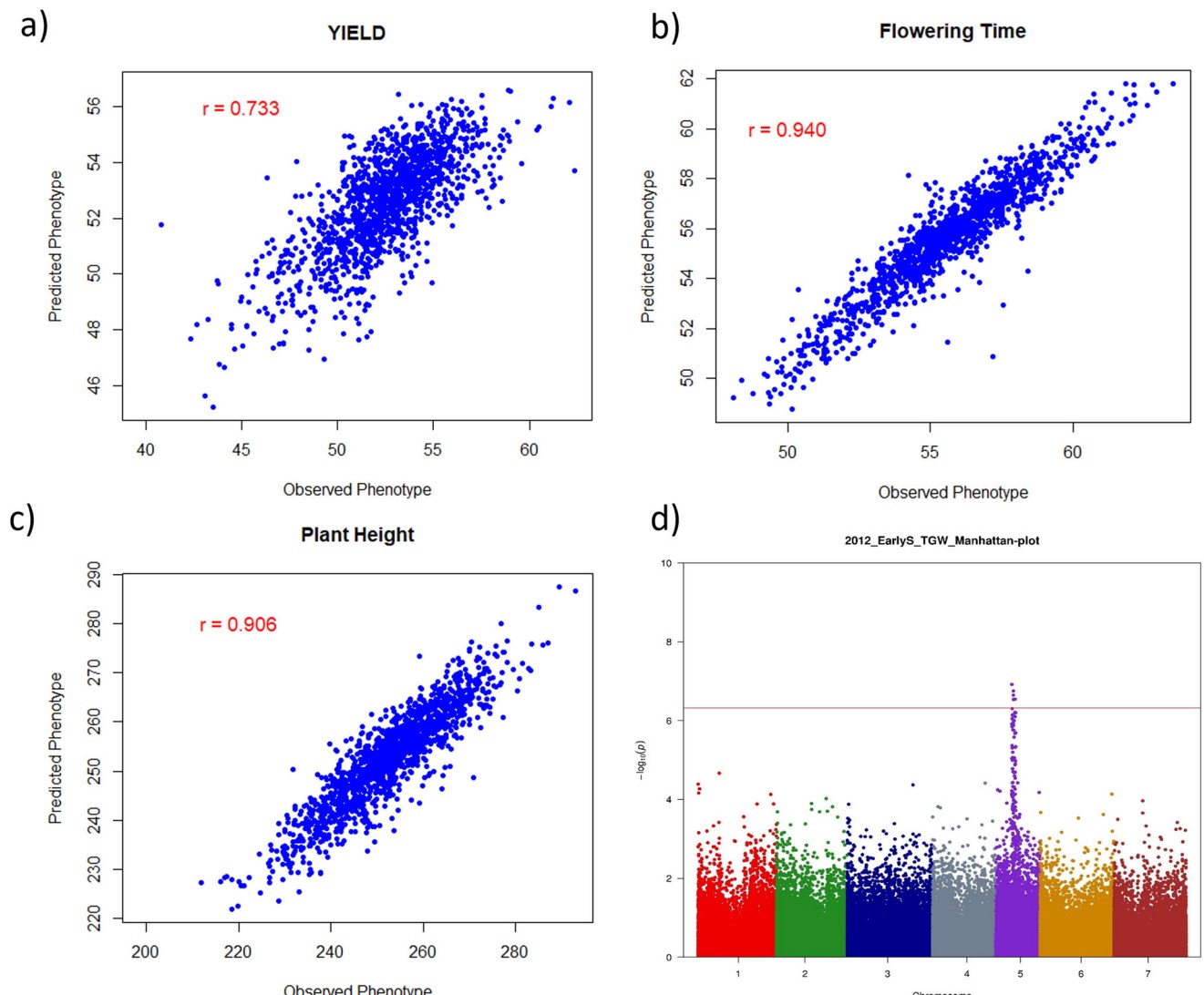

**Fig. 4 Whole genome scan with improved reference genome for genomic selection. a** Yield, **b** Flowering time, **c** Plant height, and for GWAS for test grain weight in millet (**d**).

phenotypes using association[28] and/or biparental population analyses and subsequently introgressed into elite breeding lines.

## Conclusions

The fidelity of long read sequences and improved resolution of haplotypes of the platinum-grade assemblies presented in this study provide a pangenome template that will help understanding the genomic distinctions which characterize global heterotic gene pools in pearl millet. Structural variations identified across these contrasting, heterotic pool-specific lines will complement the extensive SNP diversity repertoire in the crop and can be used to investigate the causal loci underpinning heterosis related traits such as fertility restoration, duration of pollen shed, male sterility, anthesis, maturity etc. along with agronomy and disease resistance. The sub-set of the SNPs identified here with robust PIC values between the two heterotic pools will help accelerate molecular breeding applications such as marker aided selection and whole genome prediction for a range of simple to complex traits. The comparative genome analysis presented here will also prove to be a valuable tool for genome editing applications in pearl millet in the future, especially for identifying and reintroducing agronomically important alleles that might have been lost

during domestication but conserved in wild species. Gene annotation and enrichment analyses in our study discovered that the millet genome is indeed enriched with cysteine and methionine coding genes, which agrees with the earlier finding[11] of higher concentrations of the above-said amino acids in pearl millet as compared to legumes such as pigeon peas and chickpeas. Thus, millets and legumes complement each other for balanced amino acid composition and hence the combination is recommended as balanced food to fight against malnutrition.

## Methods

**Selection of inbreds for sequencing**. In addition to improving the genome assembly quality of the existing reference genome Tift 23D$_2$B$_1$-P1-P5[4] (referred as Tift-2017), we selected additional two inbred lines that would serve as reference genomes representing each heterotic pool. Using previously published SNP data[4], we generated a kinship and diversity matrix for Tift, 843 B and ICMR 06777. While Tift was closer to the germplasm lines, 843 B was located within the B- line pool, while ICMR 06777 was in the R- line pool. These lines are also well represented in the pedigrees of the current hybrid parental lines from the Asian pearl millet

breeding programs, therefore making them ideal lines for generation of high-quality reference genomes.

**Sample preparation and library construction.** Fresh young leaf tissues of three Millet lines were collected and removed petiole and mid rib. Two grams of leaf tissue used to extract high molecular weight (HMW) DNA using Bionano-Prep Plant Tissue DNA Isolation Protocol (https://bionanogenomics.com/wp-content/uploads/2017/01/30068-Bionano-Prep-Plant-Tissue-DNA-Isolation-Protocol.pdf). Extracted nuclei were purified using density gradient method, and then nuclei suspension embedded in agarose plugs for subsequent DNA purification. The recovery of HMW DNA quantification was performed using Pippin Pulse Gel Electrophoresis & Qubit method.

**Long read sequencing.** Long-read data were generated using the Pacific BioSciences (Menlo Park, CA, USA) Sequel II platform with SMRTcells running 30-h movies. HiFi Libraries were constructed form approximately 15 μg DNA that had been sheared first by passing 20 times through a 26 G Hamilton syringe to eliminate sample viscosity and then sheared a second time using a Megaruptor 3 (Diagenode) to a narrow fragment size distribution with a center of mass at 15–20 Kbp. Fragment size distribution was confirmed using a Femto Pulse analyzer (Agilent) prior to preparation of HiFi SMRTbell Libraries using the SMRTbell Express Template Prep Kit 2.0 according to manufacturer's protocol. Fragments smaller than 15 kb were removed from the library using a 75E cassette on a Pippin HT system (Sage Science) with 15–20 kb high-pass setting. Libraries were prepared for sequencing using the Binding Kit 2.0 chemistry and sequencing was performed on Sequel II platforms per Pacific Biosciences protocol.

**Contig assembly.** HiFi reads were assembled with hifiasm (https://github.com/chhylp123/hifiasm) to generate sequence contigs with the parameter "–l0" to disable duplication purging as recommended for inbred/homozygous genomes. Primary and alternative contigs were combined to create the initial contig set. Contigs were filtered to a minimum contig length of 60–70 kb and minimum coverage varying from 5 to 20x depending on the overall coverage depth per sample to remove low coverage artifacts (Table 1). The final contig sets included 46–95 contigs per sample with contigs N50 s of 71–129 Mb (Table 1). Because hifiasm lacks a consensus polishing step, HiFi reads were aligned to the contigs using minimap2 (https://github.com/lh3/minimap2) to correct SNPs and small INDELs identified using samtools mpileup (https://github.com/samtools/samtools). QV scores for the resulting polished assemblies were calculated with Inspector (https://github.com/Maggi-Chen/Inspector) (Table 1).

**Genome mapping.** Genome maps were generated on a Bionano Genomics (San Diego, California) Saphyr platform at Corteva Agriscience™ using the Direct Label and Stain system as previously described[7]. DLE-1–labeled molecule data were filtered to create datasets with an effective molecule coverage between 61–99x and molecule N50s ranging from 353 to 443x (Table 1). Molecule datasets were assembled on the Bionano Access platform with the "non-haplotype, no extend and split, no cut CMPR" configuration. The assemblies yielded Bionano map counts of 53–80 maps with map N50s of 104–143 Mb (Table 1).

**Hybrid scaffolding.** PacBio sequence contigs were merged with Bionano genome maps to create hybrid scaffolds. Hybrid scaffolding was performed via Bionano Access with the parameters "Resolve Conflicts" and "Trim Overlapping Sequence = On" selected. Scaffolds ranging from 10-35 with scaffold N50s of 144-279 Mb were generated for the three samples (Table 1).

**Pseudomolecule construction.** The chromosome-scale pseudomolecules were built from the hybrid scaffolds using a slightly different approach for each of the three samples. Because of the high contiguity of the ICMR 06777 assembly, it was straightforward to assign chromosomes and to order and orient hybrid scaffolds relative to the Tift-2017 assembly using the previously described reference-based "scaffold-chunk"[29] method. N-gaps of 100 bp were inserted between each of the concatenated scaffolds. The remaining unscaffolded contigs were also concatenated by 100 bp N-gaps and placed in Chr00. Pseudomolecules for 843 B and Tift were built using the same approach relative to the ICMR 06777 pseudomolecules with both hybrid scaffolds and leftover contigs being incorporated.

**Omni-C validation.** The Tift pseudomolecule assembly was validated with Omni-C data from Dovetail Genomics (Scotts Valley, CA). The Omni-C library was sequenced with 100 bp Illumina (San Diego, CA) short reads on the NovaSeq platform. Paired reads (507 M pairs) were aligned to the Tift pseudomolecules using the processing pipeline provided by Dovetail (https://omni-c.readthedocs.io/en/latest/index.html). The resulting contact map was visualized with Juicebox v1.22 (https://github.com/aidenlab/Juicebox/releases) to confirm the order and orientation of scaffolds as determined by method above using ICMR 0677 as a reference.

**Genome assembly comparisons with TagDots.** TagDots is a simple and rapid approach for comparing genome assemblies and pangenome analysis. This method utilizes a set of unique sequence tags generated from each line that are aligned to all the reference genomes in the pangenome collection or to a specific line of interest. Specifically, uniquely self-mapping 300 bp sequences are selected every ~1–10 kb (depending on the desired density) to build the tag sets for each line. In this scheme, the tags from each "reference" are aligned with minimap2 to the assemblies for all pangenome assembly "queries" and the resulting uniquely mapping tag coordinates are visualized in TIBCO Spotfire® (Somerville, MA). A 100 Mb offset is applied to the coordinates of the query mappings to facilitate a unified view per chromosome.

**Comparing the assembly contiguity between genomes.** A large discrepancy was observed in the scaffold N50 between ICMR 06777 and the other two genomes (Table 1). The root cause of the reduced scaffold N50 in 843B and Tift is not immediately clear from the sequencing data metrics, such as HiFi read length and HiFi read coverage, which are typically drivers of assembly contiguity. This discrepancy is particularly notable in Tift which has longer read length and higher coverage than ICMR 06777. However, based on the relative positions of the contig/scaffold breaks in Tift and 843B compared to ICMR 06777 this lack contiguity appeared to occur near centromeric regions. To further investigate this issue, the centromeric regions were identified in ICMR 06777 by mapping occurrences of a pearl millet centromere specific satellite repeat sequence (GenBank accession: Z23007.1[30]) using BLAST (v. 2.10.0 + [31]) with a minimum $E$ value of 1e–30. Then, HiFi reads from Tift and 843B were aligned to the ICMR0677 assembly using minimap2[32]. Regions containing the centromeric satellite repeat in ICMR 06777 had significantly reduced HiFi read coverage in Tift and 843B compared to immediately adjacent regions which did not contain these repeats. HiFi sequencing is known to be capable of uniform

coverage of genomes including centromeres[33]. However, the quality and uniformity can be affected by sample quality, DNA extraction, shearing, and sample handling upstream of sequencing. Thus, the lower contiguity in these samples is partly explained by reduced coverage of reads containing centromeric repeats in those samples. In addition, centromeric regions are often characterized by a lack of Bionano map/molecule coverage due low-label DLE-1 density. Consequently, hybrid scaffolds will not be able to span centromeric region, unless the contigs themselves span these regions, which was the case for 843B and Tift, unlike ICMR 06777 which completely assembled the centromeres for 6 out of 7 chromosomes (Chr04 has scaffold gap in centromere), which had a significant positive impact on scaffold N50.

### Annotation

*Pre-annotation assembly quality assessment and repeat library construction.* Genome assembly contiguity was assessed for the three sequenced millets and Tift-2017[4] by comparing gap-density curves, genome BUSCO scores and the LTR Assembly Index (LAI). In-house developed perl scripts were used to construct a bed file of gap positions for each genome. This served as input to the R tool CMplot for the visualization of these gaps that could then be compared with each other for a visual assessment of gap prevalence in the assembly (Supplementary Fig. 1).

Another strategy employed to assess assembly quality was the comparison of genomic BUSCO scores among the four genomes. The de novo gene prediction program Fgenesh v7.2.2[34] with the "−p" setting for monocots was used on each assembly without repeat-masking, to output an annotation gff3 file. The gff3 was then used with gffread[35] to output a set of proteins for each assembly. Busco v5.1.2[14] with 4896 OrthoDB 10 Poales BUSCOs was run on each set and a comparative graph generated to assess completeness of protein-coding genes (Supplementary Fig. 2).

A custom repeat database was built for each of the three millet genomes using the EDTA pipeline[16] with parameters (--species others --anno 1 --sensitive 1 --curatedlib Poales.lib). The Poales.lib fasta file containing interspersed grass repeats, was generated from a licensed copy of REPBASE[18] and provided as curated repeats during library construction. The "--anno 1" parameter generates a masked genome file, and a repeat summary file. While the summary file was used for comparative analysis, we found that the masked genome size was inconsistent with the input genome size. Therefore, we ignored the masked genome output, and used the repeat libraries for downstream repeat masking of genomes prior to annotation. The LAI was estimated using default parameters and LTR_retriever[36] files generated as part of EDTA.

*Genome annotation.* The three millet genomes were annotated by first hard masking repeats using RepeatMasker (http://www.repeatmasker.org) and the custom repeat database. The repeat-masked genome was then used as input for gene predictors. We used a licensed version of the de novo gene prediction program Fgenesh v7.2.2[34] with parameters -full_gene -noslash -skip_prom -skip_term and -p Monocots, to conduct HMM model-based gene calling. Augustus v2.7[37] was run with default settings, and model set to "maize".

Poales EST and cDNA sequences from NCBI (6.08 million), public maize NAM line cDNA sequences from MaizeGDB (1.89 million) and monocot cDNA sequences from Phytozome (3.09 miilion) were aligned to each repeat-masked millet genome using GMAP v03-25-2018[38]. Each evidence set was run with GMAP parameters --gff3-cds=genomic --no-chimeras --cross-species --max-intronlength-middle = 7000    --max-intronlength-ends =

10,000 -n 1 --min-identity=0.60 --min-trimmed-coverage = 0.80. We also generated a total of ~8 million Pacbio IsoSeq full-length non-chimeric long reads using the PacBio Sequel II System and the IsoSeq v3 long read analysis pipeline (https://github.com/PacificBiosciences/IsoSeq). These were mapped to all millet genomes using identical GMAP parameters as earlier without the --cross-species flag and --min-identity, --min-trimmed-coverage values set to 0.85 and 0.90 respectively. In this case, GMAP was run as part of the PASA pipeline[39]. Monocot uniref90 proteins (1.50 million) were obtained from Uniprot and aligned to the genomes using SPLAN v2.1.3[40] with default settings.

Weighted consensus gene structure annotations were computed by providing all annotation files and a weighting scheme to EvidenceModeler[41]. The output was used to refine gene boundaries of PASA assembled gene models. The final PASA annotation file was combined with tRNA predictions from tRNA-ScanSE[42] and ncRNA predictions using Infernal[43] on RFAM v14.7. These tools were run in parallel to protein-coding gene annotation on each repeat-masked genome. Finally, functional descriptions of proteins were obtained using BLAST[44] against UniprotKB[45] database, and conducting an Interproscan[46] search to get Gene Ontology (GO), and PFAM domain associations. GO ids identified in the new Tift assembly were processed using the online tool Revigo[47] (http://revigo.irb.hr/) for gene enrichment analysis that summarizes them using semantic similarity measures and removes redundant GO terms and identifies the respective metabolic and molecular processes.

*Gene filtration and annotation quality.* High-confidence evidence supported gene sets were created after the genome annotation, using in-house developed bash scripts and support tools. Genes coding for the following types of peptides were deemed low quality and removed from further analysis. 1). Peptides with in-frame stop codons, 2). Peptides with length <50 amino acids, 3). Spurious peptides with hits to AntiFam[48] profiles, 4). Transposable Element related (TE-related) peptides, and 5). Ab-initio peptides without cDNA/EST support and without PFAM, Uniprot hits. TE-related proteins were discarded based on TE-related PFAM domains, similarity to Poales Repbase protein and to the TE protein library from Repeatmasker. The final set of protein-coding genes coded for peptides >50 AA, with no internal stops, no hits to the AntiFam database, no hits to transposable elements and had at least one of the following characteristics: cDNA/EST support for the gene model, and/or PFAM domians and/or Hits to UniprotKB database.

Annotation qualities were estimated by calculating descriptive statistics, BUSCO scores, Fowlkes–Mallows index, F1 measure and AED[15] scores for the filtered gene set. Genome Annotation Generator[49] and "agat_sp_statistics.pl' from the AGAT package[50] (https://doi.org/10.5281/zenodo.3552717) was used to generate simple descriptive statistics for comparing the filtered annotated GFF3's and the Public Tift annotation to each other. These included total gene number, total protein-coding genes, number of transcripts, median lengths for gene, mRNA, exon, intron, and CDS, mean transcripts per gene and the mean exons per transcript. BUSCO scores for the filtered gene sets were calculated as described earlier. Since AED is an integral part of MAKER[15] and we did not have a license to the product, we developed our own bash and perl scripts to calculate the metric. These scripts were also used to calculate two additional metrics: the Fowlkes–Mallows index (F-M index), and the F1 measure, for each final filtered transcript using the GFF3 files. Briefly, the genic features in each nucleotide and protein gff3 obtained as intermediate files during the annotation process, was coordinate compared to the final annotation file that served as the reference.

The exon sensitivity (ESn = Proportion of true exon predictions compared to the total number of correct exons including missed exon predictions) and exon specificity (ESp = Proportion of true exon predictions among all of the predicted exons including incorrectly predicted exons) was then calculated for each reference transcript with respect to every predicted or aligned transcript and protein for that locus using the tool agat_sp_sensitivity_specificity.pl from the AGAT package[50] (https://doi.org/10.5281/zenodo.3552717). The scores were then calculated as follows, for every reference transcript:

$$F1(\text{harmonic mean of ESn and ESp}) = \frac{2*(ESp*ESn)}{ESp+ESn} \quad (1)$$

$$F-M\ \text{index(geometric mean of ESn and ESp)} = \sqrt{ESp*ESn} \quad (2)$$

$$AED(\text{Annotation Edit Distance}) = 1 - \frac{ESp+ESn}{2} \quad (3)$$

All the scores for F1, F-M index, and AED for each transcript were pooled into three groups and a median calculated for each group. These were the final exon scores for each transcript.

*Genome synteny analysis and gene families.* A combined protein BLAST database with 145,406 proteins was created by taking the longest protein per gene from the three millet lines and the Public Tift genome. The All-vs-All BLAST output and a modified gff3 containing positional coordinates of each gene was given as input to the MCScanX[51] tool for calculating genome-wide synteny. The parameters used were "-s 10 -e 1e-20 -m 25" that defined a syntenic block as a continuous segment containing at least ten genes (-s 10), with the provision that a maximum of 25 non-homologous genes (-m 25) can be present between a pair of syntenic genes. To minimize cross chromosomal homologous associations, brought about by short segmental homology between proteins, the calculated $E$ value threshold was set at high specificity. An outgroup was avoided as the objective was to assess assembly quality and gene overlap in terms of syntenic blocks. The present study is not designed to assess evolution of the millet clade, so inclusion of an outgroup was unnecessary.

Protein-based gene families were constructed using the MMSeqs2 tool[52] using the following parameters "--mask 0 -c 0.8 --cov-mode 0 --min-seq-id 0.52 -e 1e-20 --max-seqs 20 --cluster-mode 1 --rescore-mode 0 --similarity-type 2 --cluster-reassign --remove-tmp-files --cluster-steps 9". An outgroup was left out because we wanted to assess the congruency in terms of gene overlap among the millet lines.

*Comparative mapping with other cereals.* Chain files that link between different assemblies are generated using FLO[53]. The minIdentity was set to 98% while other parameters set to defaults. The dot plots are generated using genome fasta files and annotation gff files which were imported in the Persephone (https://persephonesoft.com). Ortholog and synteny linking were constructed using tools provided in the Persephone. Structural variants were called using SyRi pipeline[54]. Parameter set (-c 500 -b 500 -l 100) were used for nucmer alignment stage and set (-i 90 and -l 100) were used for delta file filtering.

*Haplotype identification.* Haplotypes identification for each gene consists of two phases: input preparation and allele model construction.

In the first phase, quality control procedure is applied to raw sequencing reads to remove low quality reads and bases. Then, processed reads are aligned to the TIFT genome assembly using Bowtie2[55]. Variants are called using Samtools Mpileup program and in-house scripts with stringent parameters. Raw variants are then filtered to a high-quality set of SNPs by the criteria: 1). keep only bi-allelic SNP calls, 2). more than 60% of individuals with observed data, 3). minor allele frequency of individuals with observed data is at least 1%, 4). only 2% of individuals with observed data may carry a heterozygous genotype, and 5). neither major allele nor minor allele is a heterozygous genotype. Given high level of missing genotypes in low coverage samples, NPUTE[56] is used to impute the genotypes only at positions of high confidence. The alignment BAM files and high-quality SNP calls from the above procedure are used as input for haplotypes identification.

To identify haplotypes for a target genomic region, individuals in the sequenced population are clustered into groups using hierarchical clustering with complete linkage criterion based on the Manhattan distances among haplotypes calculated from the genotype data. For the first round of clustering, we use stringent criterion to ensure all the haplotypes within each cluster are 100% identical. For each haplotype group, the reads aligned to the target region from the individual of the same haplotypes group are then pooled into a BAM file to achieve high read coverage of the target region. For haplotype groups with the sum coverage depth greater than a given depth (30X for the Pearl Millet set), variant calling among haplotype groups is performed using pooled alignment data to create a larger set of variants than the initial variant set for the target region. The dense set of variants are then used for a second round of hierarchical clustering among haplotypes identified in the first round. The reads aligned to the target region from the individual of the same haplotypes grouped from the second round of clustering are further pooled into a BAM file. Then, Pilon[57] program is used to produce assembled haplotype sequence for each haplotype group.

For rare haplotypes (represented by less than three individuals), jumping profile Hidden Markov Model[58,59] is used to map the recombination break point(s) that generates the rare haplotype from two or more major haplotypes. Rare haplotypes with identified breakpoints have subsequences flanking the breakpoint reassigned to the relevant major haplotypes. Rare haplotype groups may not contain sufficient sequence read coverage to enable a local assembly. Segment of merged bam files of relevant major haplotype group flanking the breakpoint are further merged to create the full-length consensus sequence of the rare haplotype. Rare haplotypes lacking evidence of breakpoints may be assigned to the most likely major haplotype group to which they are mapped.

*Diversity, GWAS and hybrid prediction*
Phylogeny and principal component analysis: The SNPs were filtered to retain loci with missing proportion <20% and minor allele frequency >5% and were used in subsequent analysis. To quantify the relatedness between individuals, genetic distance matrix of all samples (PMiGAP, Parental and Wild accessions) based on identity by state was calculated using PLINK v1.90[60] with parameter "--distance". Then, neighbor-joining phylogenetic tree was constructed using "neighbor" in PHYLIP package v3.697[61] based on the distance matrix. Principal component analysis (PCA) was performed to study the relatedness and clustering among samples. The top 20 PCs of the variance-standardized relationship matrix were extracted using EIGEN-SOFT v7.2.1[62] with default parameters. The first three PCs were drawn using R (https://www.r-project.org/).

Genome-wide association study (GWAS): A pruned representative SNP set for PMiGAP accessions was generated using PLINK v1.90[60] with "--indep-pairphase 50 10 0.8" parameter. Further

GWAS analysis was carried out against phenotypic data from ref. [63] in GAPIT v3 using General Linear Model and first ten PCAs. Significant marker-trait associations were identified after applying Bonferroni correction.

Fertility restoration mapping: We recalled the SNPs using the earlier available raw RAD sequence data[4] for the 579 [maintainer (B-) and restorer (R-)] lines with 843 B as the reference genome. This resulted in a total of 209,337 SNPs (call rate 80%, MAF 5%). We mapped SNPs specific to B- line pool, and R- line pool. The analysis resulted in 7029 SNPs for the B- line pool and 40,885 SNPs for the R- line pool. These specific SNPs were annotated using the newly available platinum standard genome. The SNPs falling in genic regions were selected using currently available gene models. These were considered as specific genic SNPs (772 in B- line pool and 3351 in R- line pool). These genes were checked with for homology with the pentatricopeptide repeat *Rf* genes from sorghum (*Sorghum bicolor*)[21], sunflower (*Helianthus annuus*)[25], setaria (*Setaria viridis*)[26], and maize (*Zea mays*)[27]. A total of 14 genes were found to have a high degree of similarity (more than 95% of query coverage and percentage identity). Out of these 14 genes, 12 genes were from the R- pool while 2 genes were from the B- lines pool. Finally, we computed the sequence similarity of these genes with respect to 843 B (representing the maintainer pool) and ICMR 06777 (representing the restorer pool). Out of the 12 genes from the R- line pool, six showed 100% query coverage and percent identity while another six genes showed more than 98 percent identity score with 100% query coverage. From the B- line pool genes, the two genes showed 100% query coverage and percent identity (Supplementary Table 2). The six genes showing polymorphism (less than 100 percent identity scores) between 843 B and ICMR 06777 are the potential genes that can be targeted for fertility restoration/male sterility maintenance in pearl millet.

**Reporting summary**. Further information on research design is available in the Nature Portfolio Reporting Summary linked to this article.

## Data availability

Raw read sequences were deposited in the NCBI under accession code BioProject: PRJNA886462. Pearl millet genome assemblies and annotations are available at https://doi.org/10.6084/m9.figshare.21261129.v1. All other data included in this publication, or its Supplementary Data are also available from the corresponding author (or other resources, as applicable) on reasonable request.

## Code availability

Custom code used for genomic selection model and genome-wide data analysis is available at https://doi.org/10.6084/m9.figshare.23932707.v1.

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

## Acknowledgements

We acknowledge the Research and Development division of Corteva Agriscience™ for supporting this study. The funding support from the CGIAR Research Program on Grain Legumes and Dryland Cereals (CRP-GLDC) and the ICRISAT-Pearl Millet Hybrid Parents Research Consortium (PMHPRC) is gratefully acknowledged. This study has been published as part of the CGIAR Research Program on Grain Legumes and Dryland Cereals.

## Author contributions

R.B., H.L., M.K., A.P., R.K.V. conceived and designed the experiments; R.B., J.G., S.S., S.D., V.B., A.C., C.T.S. supervised the research; K.F. assembled the genomes; A.S., M.N. performed the annotation of the genomes; S.K.G., K.M., P.K. done sample preparation and library construction; V.L., G.M. performed the sequencing of millet lines; J.C., Y.Z., P.B., B.P-B. contributed to comparative genomics and bioinformatic analysis, R.K.S., A.M., H.G., A.R., R.G. performed diversity and GWAS analysis; P.R., R.K.S., A.S., K.F., J.C., Y.Z. and R.B. wrote the manuscript.

## Competing interests

The authors declare no competing interests.
