## [Peer Review File · Communications Biology]

Reviewers' comments:

Reviewer #1 (Remarks to the Author):

This manuscript describes thoroughly 3 new reference sequences of pearl millet. The description of the dataset is very thorough and overall clear. The genetic material used is appropriate and well described. These new references are of high quality as demonstrated. The work is original and is a very significant step-up compared to the previous reference sequence for pearl millet. I also appreciated that beyond the usual genomes comparison and quality checks the authors provide results utilizing the dataset in genetics and breeding (GWAS, GS). There is no doubt that these new datasets will make significant differences in genetics and breeding of pearl millet. The manuscript is overall well written and figures are well designed and clear. I however made some suggestions to improve the manuscript further in the attached pdf files.

Reviewer #2 (Remarks to the Author):

General overview

The study, "Improved pearl millet reference genome assemblies representing global heterotic pool provide a reliable genomic resource for molecular breeding applications" by Ramu et al., present three new high quality genome assembly for pearl millet and used them for comparative and preliminary GWAS analysis of agronomically important traits. The results highlighted major improvement of the genome assembly at multiple levels, structural, gene and repetitive sequence coverage. GWAS allowed to identify QTL multiple traits including for grain weight.

Comments:

- I find the manuscript well written and genome assembly and comparative analysis very robust. However the manuscript has no subheading for any of the paragraphs, it should be structured in subsections. As it is now it gave me the impression that this was submitted as a short communication, but in the system it indicate that was submitted as a research article. The paper would need a more in depth investigation and presentation of the GWAS, candidate genes analysis.
- The GWAS and haplotype analysis appear to be very descriptive, and preliminary analysis with no much connection to each other;
- Most of the GWAS analysis appear to be preliminary results, the section could be written as a separate section and linked to the haplotype analysis, to provide an example of how haplotype can help to narrow number o candidate genes;
- In several cases the author refer to the term pangenome. I would suggest to refer to comparative analysis and the results can highlight the need to develop a pangenome for pearl millet that typically include a larger number of samples representing most of the genes present in a given species.

Reviewers #3-4 (co-reviewers) (Remarks to the Author):

Reviewer summary

This manuscript describes the work to create 3 high-quality reference assemblies in the important crop pearl millet. From the described methods and analytics, the authors have achieved a meaningful result in higher-quality assembly for these three lines. They show the value and importance of generating high quality genomes by showing improved outcomes that the community could have through haplotype descriptions, GWAS mapping, and fertility restoration examples. Overall, this manuscript delivers a valuable resources to the community via new reference assemblies and refined variation and haplotypes. This manuscript should be accepted but with some moderate to significant revisions and responses to certain issues.

General notes:

Is the manuscript broken down into sections? There are no labels for Intro/Results/Methods. The methods and results section are somewhat intermingled.

Please check for continuity in spelling and typos in the document. An example is "Fig. 2" vs "Fig 2" on line 103 and 105, respectively. Supplementary Figure captions are another.

Many of the data files do not seem to be available for reviewer evaluation (SNP, GFF, haplotypes, germplasm list, etc). Please provide these or state if they are publicly deposited elsewhere.

Specific notes:

Line 41-53. The authors should consider mentioning the issues of fertility restoration here since it is addressed at the end of the Results section. This is a valuable tool within breeding and highlighting it in the introduction could strengthen the narrative a bit.

Line 45. Not sure if this was meant to read "economies" or "ecologies". Are you trying to refer to ecological areas that are of sufficient quality to grow major cash crops?

Line 63: "were carefully selected to represent the global heterotic pattern as established by the ICRISAT pearl millet breeding program". The authors specify these criteria in the Methods (Line 275) and they are good criteria for incorporation, but they also mention a PCA kinship matrix. I am not clear which SNP PCA they are referring to in the supplemental data or if it is in this manuscript at all (Figure 3, Supplemental Figure 10, or 11). Please specify by denoting it in the Methods section or at Line 63. If it is figure 3, supplemental figure 10, or 11, please also highlight the location of the three lines within the PCA generated from the kinship matrix.

Line 65: Seems like this is where the "Results" section header should be.

Line 70: Could the authors elaborate on the large discrepancy in scaffold N50 between ICMR 06777 and the other two genomes? Coverage? Size of inserts. Also there is a difference in the number of genes models between the lines.

Line 84: what makes the centromeric region of ICMR 06777 more amenable to HiFi read-through or contig assembly? Was it the longer read length, as the coverage was lower, is this biology?

Line 96: Is there drought phenotype information available for the 3 cultivars? Any difference between ICMR 06777 and the other genomes lacking the chromosome 4 inversion?

Line 108: Does this line suggest the assembly is incomplete, or the genome has lost the genes? If you look between the other two assemblies is the synteny conserved.

Line 111: What is the ensemble-based approach?

Line 123: The written "ICMR06777" doesn't have a space in between the ICMR and 06777 as in the rest of the paper. This also occurs at Line 324.

Line 129 hints at the 'novel' genes discovered in this analysis could be due to the sulphur-containing AAs and metabolically-dependent micronutrients (Fe and Zn). While this is interesting, these genes are novel relative to the prior assembly from 2017.

A stronger argument for this section's inclusion in the manuscript would be a comparison to these gene homology in pearl millet relative to others in panicoideae or andropogoneae? Something akin to

what you did for the QTL portion of the study starting at Line 181. Were these genes in regions of the assembly, missing from the BACs, were these regions that were hard to assembly and represent regions that were collapsed in the previous assembly.

Line 129 (and Supplemental Figure 4). How was the GO enrichment done (Bonferroni corrected, something else)? Or was this just a distribution/list of GO terms associated with the genes. I can't find the methods write up specifically for this.

Line 164: The authors should redo the analysis with the current genome version for BTx623 (v3.1), B73 (v5), Nipponbare (IRGSP v1), and foxtail millet (v2). I realize this shouldn't change the structural comparison much, but the genomes used in the manuscript are quite old.

Line 166: Please explain further how the dual alignment of B73 and Tift revealed the ancient maize WGD event.

Line 138: BUSCO scores should be follow the quality, and then discuss the biology. BUSCO scores of 92% are still low, compared to the new assemblies, suggesting they are still missing information.

Line 167: Need to include the reference to the whole duplications previously demonstrated, need to consider what reference to include.

Line 172: Consider providing a VCF of variation data used in the analyze 646K set.

Line 176: Is this data publicly available for the breeding program material?

Line 181: Are the germplasm phenotypes available?

Line 183. Can the authors discuss how this GWAS is an improvement over using the 2017 alignment? I assume it's significantly better, but a quick description of prior trait discovery compared to this improved reference would be a nice demonstration. That being said, I am not asking for the authors to redo this GWAS using the 2017 alignment for an additional figure.

Line 193: Are the haplotypes available?

Line 201: is where the "Discussion" Header should go.

Line 203: Typo/grammar: Pearl Millet is capitalized here but not in rest of manuscript. Should also read "heterotic gene pools in the pearl millet crop" or read "heterotic gene pools in pearl millet".

Line 207: What are PIC values? Is the table of these markers available for public/academic breeders, or only to ICRISAT? Corteva?

Line 201-216: The discussion section seems rather short. It could be expanded to include descriptions about additional content b/w the millet germplasm: varying TEs; where the TEs are relative to the inversions; singleton gene content in ICMR 06777 vs other lines, ncRNA content, etc. This is not necessary, but I think it would improve the section a bit.

Line 253: Figure 3 panel B. Reference point(s) are difficult to identify in PCA. Perhaps highlight with an arrow or use a more contrasting color to the blue for the A line.

Line 265: 'for GWAS' is in bold compared to rest of caption.

Line 361: Are they able to share this repeat masking database?

Figures/Tables

Table 1: median RNA length differs by >100 bp for 843B and the others, but the median exon and intron lengths.

Supplemental Figure 1. While it is stated in the caption, the authors should label the legend and change it so that it is clear the gap values are x102.

The caption for Supplementary Figure 3 is labeled as "Supplementary Figure 2".

The caption for Supplementary figure 5 is labeled as "Supplementary Figure 4".

Supplemental Figure 7 caption needs a period at the end of the sentence.

Responses to reviewer's comments

Reviewer 1:

Reviewer's comment: Line 55 Is it an inbred line? Then say it (instead of germplasm line).

Response: Tift23D₂B₁-P1-P5 is an inbred line. We have changed from 'germplasm line' to 'inbred line' in the main text file.

Reviewer's comment: Line 63 "global heterotic pool pattern". I'm not sure what it means. Do the 3 lines represent 3 heterotic pools commonly used? If so, do they represent different quality or marker types of millet? Or different continents?

Response: ICRISAT is one of the oldest institutions that initiated research work on hybrid development in Pearl Millet long time ago and established primarily two heterotic pools namely, male (R-lines) and female (B-lines) based on the trait-specific breeding which has happened for separate set of traits in seed and restorer parents in last about 50 years. ICRISAT runs a pearl millet improvement consortium wherein the potential parental lines are distributed to member companies/organizations and hence most of the millet germplasm prevalent in the current day owe their origin to ICRISAT's founder germplasm. Tift line was chosen in the current study as it is one of the oldest lines as well as the one on which original genome sequencing was conducted earlier by Rajeev et al (2017). The other two lines included in this study are carefully chosen by the breeders to represent the characteristics present in R and B pool to represent the active breeding germplasm that is in use currently across the world for hybrid development.

Reviewer's comment: Line 79 That's a strange expression and unnecessary complicated. A good reference isn't necessary for GS (that's the beauty of GS) so I would rather say: "particularly for pangenomic, haplotype-based and introgression breeding."

Authors response: Reviewer's suggestion is included.

Reviewer's comment: Line 83 Table 1 says 6432 gaps in Tift-2017 and 25 gaps in Tift, so?

Authors response: We thank reviewers for catching this. We have corrected these numbers in the main text.

Reviewer's comment: Line 85 How do you know it matches the centromeres? What data did you use to identify the centromere?

Authors response: Now we have included a paragraph in methods section on identifying the centromeres in the genome. 'Putative centromeric regions are often readily identifiable in the BioNano hybrid scaffolds by low label density in the PacBio contigs and a corresponding lack of map coverage due to low label density in the molecules. In addition, the centromeres were identified by mapping *P. glaucum* satellite DNA to the assemblies. This centromere specific sequence does not contain DLE-1 labels.'

Reviewer's comment: Line 86 Why PacBio gave better results in ICMR? The same technology was used for other lines also, right?

Authors response: Even through all genomes assembled with the same technology, the root cause for difference in is not immediately clear from the sequencing input data metrics, particularly in the case of Tift which has higher coverage and longer reads (Table1). It is observed that ICMR has more HiFi read coverage in centromeric repeat regions compared to Tift and 843B. However, the quality and uniformity are affected by sample quality, DNA extraction, shearing, and sample handling upstream of sequencing.

Reviewer's comment: Line 108 I don't really understand your point here. If Tift assembly is replacing the old Tift-2017, I don't see a point of discussing the synteny with Tift-2017.

I strongly suggest you separate the two points in different sections:

1- you compare the quality of Tift and Tift-2017

2- you discuss the synteny among 843 B, ICMR 06777 and Tift (but not Tift-2017)

Authors response: We completely agree with reviewer's suggestions. We removed this sentence and changed the manuscript in several places to reflect the reviewer's comment.

Reviewer's comment: Line 112 Sentence grammatically incorrect (The analysis produced a set of genes were annotated"). Please rewrite.

Authors response: Thank you reviewer for pointing out this grammatical mistake. We rewrote the sentence in the main text.

Reviewer's comment: Line 136 I don't know what the 'higher micronutrient' is? I guess you mean higher concentration

Authors response: Yes, we agree with reviewer's comment. We mean higher concentrations of micronutrients here. Changes were made in main text.

Reviewer's comment: (line 166) I don't understand what you mean here. Do you mean "for most Tift sequences"?

Authors response: Yes, we mean here 'the most of Tift sequences. Corrected in the main text.

Reviewer's comment: (line 169) If you want a broader audience to understand (may be breeders) I suggest to add "as shown by the contiguous lines in Supplementary 9b etc..' or something like this.

Authors response: Thank you to the reviewer for the suggestion. Sentence was rewritten as per reviewers' suggestion.

Reviewer's comment (line 179): I would emphasize this point by separating into two sentences. .. in whole genome prediction approaches to create the hybrids. The prediction accuracy was very high from 0.73 to...

Authors response: We agree with the reviewer and have made the appropriate changes.

Reviewer's comment (line 215): Do the end product significantly enriched in cysteine and methionine? I think this last statement on smart food is bit bold when based only on gene annotation without measurement of cysteine and methionine in grains.

Authors response: While the scope of current study didn't allow us to analyze the amino acid/protein profiling of millet grains, it is indeed very well established by our own collaborators in the past (Anita et al. 2019) that millets are indeed significantly enriched for Cysteine and Methionine content, which are generally deficient in legumes. The traditional food combination patterns followed in many rural areas of India (Millets + Legumes; "Bajra ki Roti-Dal") is indeed a testament to the conventional food consumption wisdom that millets and legumes complement each other for balanced amino acid composition; the current study has only reinforced the already known/established protein composition of millets in the form of genomic evidences and in our opinion 'smart food' is just a modern day rebranding of age-old wisdom to address the menace of micronutrient malnutrition!

Reviewer's comment (line 245 – Fig 1 legend): How did you determine the position of centromere? Please indicate where the centromere is on the graph.

Authors response: A separate paragraph is now added to the methods section about determining the centromeres. 'Putative centromeric regions are often readily identifiable in the BioNano hybrid scaffolds by low label density in the PacBio contigs and a corresponding lack of map coverage due to low label density in the molecules. In addition, the centromeres were identified by mapping *P. glaucum* satellite DNA to the assemblies. This centromere specific sequence does not contain DLE-1 labels. We have changed to 'pericentric region' from 'centromere region' because the assembly improvement spans a much wider region.'

Reviewer's comment: Please use the same style for the paragraph across.

Authors response: Yes, we followed the reviewer's suggestions and followed the same style across the manuscript

Reviewer's comment (line 273): I think this information about the 2 inbred lines should be in the main text (representatives of maintainer and restore pools).

Authors response: As per reviewer's suggestion, information describing about 2 inbred lines is now added to the main text.

Reviewer's comment (line 283): HMW, full name?

Authors response: Full name of HMW (high molecular weight) is added to the main text at the first occurrence.

Reviewer's comment (line 286): 2nd instance, use the acronym, HMW

Authors response: Acronym was added to the main text at the 2nd instance.

Reviewer's comment (line 304 and 305): Correct as Table 1, instead of Table1

Authors response: Corrected in the main text and the same style is followed across the manuscript.

Reviewer's comment (line 348): Remove capital letters for title all through the document.

Authors response: Removed the capital letters throughout the documents for all the titles.

Reviewer 2:

Reviewer's comment: I find the manuscript well written and genome assembly and comparative analysis very robust. However, the manuscript has no subheading for any of the paragraphs, it should be structured in subsections. As it is now it gave me the impression that this was submitted as a short communication, but in the system, it indicates that was submitted as a research article. The paper would need a more in-depth investigation and presentation of the GWAS, candidate genes analysis.

Authors response: Agree with the reviewer's comments and the manuscript is now presented in sections.

This article is mainly focused on improving the assembly and annotations. Limited GWAS findings and their details are presented in the current study as it was mostly to demonstrate the utility of improved genome assembly rather than discovering novel genomic regions responsible for important agronomic and other traits. A detailed manuscript is in fact being prepared with GWAS findings as the principal focus.

Reviewer's comment: The GWAS and haplotype analysis appear to be very descriptive, and preliminary analysis with no much connection to each other. Most of the GWAS analysis appear to be preliminary results, the section could be written as a separate section and linked to the haplotype analysis, to provide an example of how haplotype can help to narrow number to candidate genes.

Authors response: As mentioned above, the major objective of this paper is to provide the improved genome assemblies of key founder lines, which could be used in variety of ways in down-stream applications such as GWAS, diversity, candidate gene and haplotype analyses. We have provided the haplotypes information for genes in improved Tift so that interested breeders can explore the available haplotypic variation for a given locus across the germplasm used in the current study for any further targeted utilization.

Reviewer's comment: In several cases the author refers to the term pangenome. I would suggest referring to comparative analysis and the results can highlight the need to develop a pangenome for pearl millet that typically include a larger number of samples representing most of the genes present in a given species.

Authors response: We completely agree with the reviewer's suggestion. We changed from 'Pangenome' to comparative analysis wherever it is applicable.

Reviewer 3:

General notes:

Reviewer's comment: Is the manuscript broken down into sections? There are no labels for Intro/Results/Methods. The methods and results section are somewhat intermingled

Authors response: Agree with the reviewer's comments and the manuscript is now presented in sections.

Reviewer's comment: Please check for continuity in spelling and typos in the document. An example is "Fig. 2" vs "Fig 2" on line 103 and 105, respectively. Supplementary Figure captions are another.

Authors response: Thank you reviewers for catching this typo. We have thoroughly checked and followed the same format throughout the manuscript.

Reviewer's comment: Many of the data files do not seem to be available for reviewer evaluation (SNP, GFF, haplotypes, germplasm list, etc). Please provide these or state if they are publicly deposited elsewhere

Authors response: GFF and assembly were already shared, and link is provided in data availability section. The list of germplasm is the same as available in Varshney et al. 2017 paper, which we cite in the manuscript. SNP data and haplotypes information is now made available for public now under the same link as provided in data availability section.

Specific notes:

Reviewer's comment: Line 41-53. The authors should consider mentioning the issues of fertility restoration here since it is addressed at the end of the Results section. This is a valuable tool within breeding and highlighting it in the introduction could strengthen the narrative a bit.

Authors response: We agree with the reviewer's suggestion. As discussed earlier with reference to the other reviewers' comments, the major focus of this study was to make available improved genome assembly and hence we kept the focus confined to that aspect; wherever possible, we have highlighted how the improved assembly could be utilized in the down-stream applications. Fertility restoration is indeed an important aspect in hybrid development and this improved assembly can unravel the genetics of fertility restoration, which is highlighted in the latter part of the manuscript.

Reviewer's comment: Line 45. Not sure if this was meant to read “economies” or “ecologies”. Are you trying to refer to ecological areas that are of sufficient quality to grow major cash crops?

Authors response: We mean here the ‘ecologies’ – referring to environmental preferences to grow major crops like maize, rice, and wheat.

Reviewer's comment: Line 63: “were carefully selected to represent the global heterotic pattern as established by the ICRISAT pearl millet breeding program”. The authors specify these criteria in the Methods (Line 275) and they are good criteria for incorporation, but they also mention a PCA kinship matrix. I am not clear which SNP PCA they are referring to in the supplemental data or if it is in this manuscript at all (Figure 3, Supplemental Figure 10, or 11). Please specify by denoting it in the Methods section or at Line 63. If it is figure 3, supplemental figure 10, or 11, please also highlight the location of the three lines within the PCA generated from the kinship matrix.

Authors response: Line 63 refers to the analysis done using the SNP data already published at Varshney et al (2017) on Tift-2017 genome (old) to identify the representative samples for genome assembly and annotation. Filtering methods for this SNP data is available in Varshney et al (2017) methods section. At this line, we are not referring to Figure 3 or Supplementary Figure 10.

Figure 3 and Supplementary Figure 10 refers to the same germplasm used in Varshney et al (2017), but SNP data used to generate these Figure 3 and Supplementary Figure 10 are called using new reference genomes described in this paper.

We did highlight the selected 3 lines in PCA plot but the 2D nature of the plot made this hard to discern.

Reviewer's comment: Line 65: Seems like this is where the “Results” section header should be.

Authors response: The manuscript is now divided into sections to assist in reader understanding.

Reviewer's comment:

Line 70: Could the authors elaborate on the large discrepancy in scaffold N50 between ICMR 06777 and the other two genomes? Coverage? Size of inserts. Also there is a difference in the number of genes models between the lines.

Line 84: what makes the centromeric region of ICMR 06777 more amenable to HiFi read-through or contig assembly? Was it the longer read length, as the coverage was lower, is this biology?

Authors response: Now we have included a paragraph in methods section on large discrepancy in scaffold N50 of ICMR and other genomes. ‘The root cause of the large discrepancy in scaffold N50 between ICMR 06777 and the other two genomes is not immediately clear from the sequencing input data metrics, particularly in the case of Tift which has higher coverage and longer reads (Table1). However, based on the relative positions of the contig/scaffold breaks in Tift and 843B compared to ICMR 06777 this lack of contiguity appears to occur in the centromeric regions. To further investigate this issue, HiFi reads from Tift and 843B were aligned to the ICMR06777 assembly. Regions containing the centromeric satellite repeat in ICMR 06777 had significantly reduced HiFi read coverage in Tift and 843B compared to immediately adjacent regions which did not. Lower contiguity in these samples is explained by reduced coverage of centromeric repeats in those samples. HiFi sequencing is known to be capable of uniform coverage of genomes including centromeres (Nurk 2022 Figure 3). However, the quality and

uniformity is affected by sample quality, DNA extraction, shearing, and sample handling upstream of sequencing. Modern protocols (SMRTbell prep kit 3.0) aim to improve consistency, but some performance issues remain difficult to predict before sequencing.'

Several factors can contribute to gene model number differences. Size of genome, interspersed repeat associated gene models, SNP's and indels at or near splice junctions, PAV and CNV among the lines. In addition to this, any automated annotation will always call protein-coding non-TE related gene models that are poorly supported with evidence. This is mostly true for *ab-initio* predicted models, leading to slight bloating of gene models. However, if you look at the protein-coding non-TE related gene loci numbers in the manuscript, they are very close between 35000 – 38000.

Reviewer's comment: Line 96: Is there drought phenotype information available for the 3 cultivars? Any difference between ICMR 06777 and the other genomes lacking the chromosome 4 inversion?

Authors response: The three cultivars were selected based on gene pool representation. ICMR 06777 represents the restorer (R-) gene pool while 843B represents the maintainer (B-) gene pool of the current breeding programs. Tift23D2B1-P1-P5 is a historical founder line. Therefore, the differences between the three lines would not stem from their drought adaptation reaction, but the gene pools they represent. We did not generate any data on drought tolerance on this germplasm.

This manuscript mainly focussed on providing platinum-quality reference genomes to the research community for further downstream analysis like investigating at the inversions between these lines to find whether they have any significant difference at these inversions or not.

Reviewer's comment: Line 108: Does this line suggest the assembly is incomplete, or the genome has lost the genes? If you look between the other two assemblies is the synteny conserved.

Authors response: Reviewer 1 also has the same comment and suggested not to compare synteny with old reference assembly, Tift-2017. And we have included the comparisons among new assemblies.

Reviewer's comment: Line 111: What is the ensemble-based approach?

Authors response: A hybrid approach comprising of *ab-initio* predictors and evidence-based gene callers. We have removed the term 'ensemble' and called it as 'hybrid approach' in the main text for better understanding.

Reviewer's comment: Line 123: The written "ICMR06777" doesn't have a space in between the ICMR and 06777 as in the rest of the paper. This also occurs at Line 324.

Authors response: Thank you reviewers for catching this typo. We have thoroughly checked and followed the same format throughout the manuscript.

Reviewer's comment: Line 129 hints at the 'novel' genes discovered in this analysis could be due to the sulphur-containing AAs and metabolically-dependent micronutrients (Fe and Zn). While this is interesting, these genes are novel relative to the prior assembly from 2017.

A stronger argument for this sections inclusion in the manuscript would be a comparison to these gene homology in pearl millet relative to others in panicoideae or andropogoneae? Something akin to what you did for the QTL portion of the study starting at Line 181. Were these genes in regions of the assembly, missing from the BACs, were these regions that were hard to assembly and represent regions that were collapsed in the previous assembly.

Authors response: We agree with the reviewer's comments. We strongly believe that technology advances using Pac-Bio long read sequences might have captured these novel sequences which were hard to capture and assemble using small read sequences. We thank the reviewer for the suggestion on detailed comparison between millet and other genomes, which we will certainly pursue as a subsequent study and attempt to publish those findings in future.

Reviewer's comment: Line 129 (and Supplemental Figure 4). How was the GO enrichment done (Bonferroni corrected, something else)? Or was this just a distribution/list of GO terms associated with the genes. I can't find the methods write up specifically for this.

Authors response: Now we have added the process describing GO enrichment analysis to methods section. 'GO enrichment analysis was done using online tool called REVIGO which summarizes them using semantic similarity measures and removes redundant GO terms and identifies the respective metabolic and molecular processes.'

Reviewer's comment: Line 164: The authors should redo the analysis with the current genome version for BTx623 (v3.1), B73 (v5), Nipponbare (IRGSP v1), and foxtail millet (v2). I realize this shouldn't change the structural comparison much, but the genomes used in the manuscript are quite old.

Authors response: We redid the analysis of comparative genome mapping of pearl millet genome with latest versions of maize (B73 v5), sorghum (BTx623 v3.1) and rice (Nipponbare, IRGSP v1). We did not see many differences between old and new genome versions. New figures were added to the Supplementary Figure 9. We kept Foxtail millet v1 comparative maps only, because the v2, genome is not assembled at the chromosome level. For example, v2 contains 436 scaffolds which made it hard to visualize the comparisons whereas v1 contains 9 pseudomolecules and easy to visualize the comparisons

Reviewer's comment: Line 166: Please explain further how the dual alignment of B73 and Tift revealed the ancient maize WGD event

Authors response: Dual alignment revealed a duplication event in maize. Please refer to Supplementary Figure 9. For example.: chromosome 6 of pearl millet has 2 copies in maize genome (chromosome 4 and chromosome 9). Similarly, chromosome 7 of pearl millet has 2 copies in maize genome (chromosome 3 and chromosome 8).

Reviewer's comment: Line 138: BUSCO scores should be follow the quality, and then discuss the biology. BUSCO scores of 92% are still low, compared to the new assemblies, suggesting they are still missing information.

Authors response: Compared to the Tift-2017 version with a completion of 78%, it's a huge improvement. Further, while BUSCO's are a good reference to check quality, its only as good as the data

used to construct the BUSCO's. In other words, bonafide novel genes, or any deletions of BUSCO genes or variants of existing BUSCO genes from highly variant taxonomic lineages like the one pearl millet belongs to, will be missed. The Poales ODB10 is created from a diverse group, and the only species closest to Millet is Setaria that is diverged for over 22 MYA (Zhang et. Al. 2022 - <https://doi.org/10.1111/1755-0998.13612>). It is likely that BUSCO HMM's will miss certain genes. At the same time, the present annotation is far from being comprehensive and this is out of scope for the present study.

Reviewer's comment: Line 167: Need to include the reference to the whole duplications previously demonstrated, need to consider what reference to include.

Authors response: Reference is included referring maize whole genome duplications (Schnable et al. 2011)

Reviewer's comment: Line 172: Consider providing a VCF of variation data used in the analyze 646K set.

Authors response: Complete SNP data is made available now to public through the same link provided in data availability section.

Reviewer's comment: Line 176: Is this data publicly available for the breeding program material?

Authors response: This material is available for public use from ICRISAT. User can request this germplasm material by filling Standard Material Transfer Agreement (SMTA) from ICRISAT.

Reviewer's comment: Line 181: Are the germplasm phenotypes available?

Authors response: Yes, they are available. They are essentially the same drought phenotypic data sets that were used to carry out GWAS analyses in Varshney et al., 2017. Nat. Biotech. Data generated at Corteva on downy mildew is not available to public.

Reviewer's comment: Line 183. Can the authors discuss how this GWAS is an improvement over using the 2017 alignment? I assume it's significantly better, but a quick description of prior trait discovery compared to this improved reference would be a nice demonstration. That being said, I am not asking for the authors to redo this GWAS using the 2017 alignment for an additional figure.

Authors response: While publishing the previous genome, Varshney et al. 2017 did GWAS analysis using association panel included in 1008 germplasm set. In the current study, we have re-called the SNPs from the same germplasm (pearl millet association mapping panel (PMiGAP)) using the platinum-standard reference genome. The called SNPs were subject to standard filtration criteria to generate an improved HapMap used for the current GWAS analysis. The previous study indeed had large gaps and hence didn't represent all the genomic regions uniformly whereas the current one had better genomic coverage which is evident from a number of marker-trait associations revealed for several traits.

Reviewer's comment: Line 193: Are the haplotypes available?

Authors response: Haplotypes data is now made available now to public through the same link provided in data availability section.

Reviewer's comment: Line 201: is where the "Discussion" Header should go.

Authors response: The article is now subdivided into various sections.

Reviewer's comment: Line 203: Typo/grammar: Pearl Millet is capitalized here but not in rest of manuscript. Should also read "heterotic gene pools in the pearl millet crop" or read "heterotic gene pools in pearl millet".

Authors response: Thank you reviewer for catching the typo. It is corrected in main text.

Reviewer's comment: Line 207: What are PIC values? Is the table of these markers available for public/academic breeders, or only to ICRISAT? Corteva?

Authors response: Now, we have made the complete SNP data/markers data available through the data availability section to the public to explore further. PIC stands for Polymorphic Information Content. The PIC values showed a significant range. These were used to develop SNP-based QA/QC panel and mid-density panel. However, these are not part of this study. We are planning to publish this data soon and once these markers added to a genotyping platform, genotyping platform will be made available to public to use them in their breeding program.

Reviewer's comment: Line 201-216: The discussion section seems rather short. It could be expanded to include descriptions about additional content b/w the millet germplasm: varying TEs; where the TEs are relative to the inversions; singleton gene content in ICMR 06777 vs other lines, ncRNA content, etc. This is not necessary, but I think it would improve the section a bit.

Authors response: We did consider this suggestion but couldn't gain consensus among the co-authors as some felt it would be over-stretching without commensurate evidence through additional experiments/analyses.

Reviewer's comment: Line 253: Figure 3 panel B. Reference point(s) are difficult to identify in PCA. Perhaps highlight with an arrow or use a more contrasting colour to the blue for the A line.

Authors response: Now the reference point is added by an arrow and text in Figure 3. Now we can easily differentiate the reference genome with the rest of inbred lines.

Reviewer's comment: Line 265: 'for GWAS' is in bold compared to rest of caption.

Authors response: Bold letters removed and corrected in the main text

Reviewer's comment: Line 361: Are they able to share this repeat masking database?

Authors response: Unfortunately, this database was generated from a licensed copy of REPBASE, and hence governed by proprietary/legal clauses.

Figures/Tables

Reviewer's comment: Table 1: median RNA length differs by >100 bp for 843B and the others, but the median exon and intron lengths.

Authors response: Thank you reviewers catching this. These are average values rather than median values. Now it has been corrected in Table 1.

Reviewer's comment: Supplemental Figure 1. While it is stated in the caption, the authors should label the legend and change it so that it is clear the gap values are x102

Authors response: We have changed the legend and replaced in Supplementary Figure 1.

Reviewer's comment: The caption for Supplementary Figure 3 is labelled as "Supplementary Figure 2".

Authors response: Thanks for the reviewer for catching this error. Corrected in the Supplementary File.

Reviewer's comment: The caption for Supplementary figure 5 is labelled as "Supplementary Figure 4".

Authors response: Thanks for the reviewer for catching this error. Corrected in the Supplementary File.

Reviewer's comment: Supplemental Figure 7 caption needs a period at the end of the sentence.

Authors response: Added the period after the end of sentence for Supplementary Figure 7.

REVIEWERS' COMMENTS:

Reviewer #2 (Remarks to the Author):

Dear authors and editor

I appreciate the effort made by the authors to address the comments. However, the authors have not addressed my major comments about the paper been very descriptive of the new genome assemblies and not looking at any novel biological or evolutionary question in more details. Recent similar papers published in this journal do focus on biological and/or evolutionary questions: here are a few examples:

- <https://www.nature.com/articles/s42003-023-04980-2>
- <https://www.nature.com/articles/s42003-022-04229-4>
- <https://www.nature.com/articles/s42003-023-04949-1>

there are much more examples that the authors can easily find in the journal using the search tab. There are specialized sections of some journals (e.g. G3) that publish new improved genome assemblies or specialized journals like nature Scientific Data <https://www.nature.com/sdata/principles>. This would be more appropriate options for this type of data.

Reviewer #3 (Remarks to the Author):

I thank the authors for their rebuttals and changes to the document. I think this meaningfully improves the manuscript and I believe it should be published (with some minor typo fixes). Apologies for missing the annotation file locations in the Figshare link that was initially provided.

Responses to reviewer's comments

Reviewer 2:

Reviewer's comment: I appreciate the effort made by the authors to address the comments. However, the authors have not addressed my major comments about the paper been very descriptive of the new genome assemblies and not looking at any novel biological or evolutionary question in more details. Recent similar papers published in this journal do focus on biological and/or evolutionary questions: here are a few examples:

- <https://www.nature.com/articles/s42003-023-04980-2>
- <https://www.nature.com/articles/s42003-022-04229-4>
- <https://www.nature.com/articles/s42003-023-04949-1>

there are much more examples that the authors can easily find in the journal using the search tab. There are specialized sections of some journals (e.g. G3) that publish new improved genome assemblies or specialized journals like nature Scientific Data <https://www.nature.com/sdata/principles> . This would be more appropriate options for this type of data.

Authors response: Thank you the reviewer for suggesting valuable papers to address biological/evolutionary questions. In response, explored the possibility of highlighting an evolutionary insight based on the findings from this study. Of the several results that were reconsidered, we did find a clear connection for the high frequency of singletons in one of the three genomes with a possible origin of geographic expansion, supporting West African origin of cultivated pearl millet germplasm. Thanks to the reviewer for this contribution!

we have added the following sentences to the main manuscript and are highlighted in yellow.

'Singletons are rare variants that offer insights into past demographic events in a species, when studied across geographical space. By simulating the empirical distribution of singletons, Cubry et al (2017) obtained results supporting West Africa (Northern Mali) as most probable origin of geographic expansion for the cultivated species of Pearl Millet. Interestingly, the higher frequency of singletons as discovered in the current study in ICMR 06777 corroborates the above finding as the line is derived from the founder population, MC94 which is a mixture of Inari landraces from Togo, Ghana and other West African countries.'

Reviewer 3:

Reviewer's comment: I thank the authors for their rebuttals and changes to the document. I think this meaningfully improves the manuscript and I believe it should be published (with some minor typo fixes). Apologies for missing the annotation file locations in the Figshare link that was initially provided.

Authors response: We thank the reviewer for his/her valuable suggestions to improve the manuscript. Typos were fixed.